# Relationship Alignment for View-aware Multi-view Clustering

**Shuangmei Peng**[1,2], **Zhe Chen**[1,2]*, **Tianyang Xu**[3], **Xiao-Jun Wu**[3]

[1] Anhui University of Technology, Ma'anshan, 243032, Anhui, China
[2] Anhui Province Key Laboratory of Digital Twin Technology in Metallurgical Industry, Anhui, China
[3] Jiangnan University, Wuxi, 214122, Jiangsu, China
{psm,cz}@ahut.edu.cn, {tianyang.xu,wu_xiaojun}@jiangnan.edu.cn

## Abstract

Multi-view clustering improves clustering performance by integrating complementary information from multiple views. However, existing methods often suffer from two limitations: i) the neglect of preserving sample neighborhood structures, which weakens the consistency of inter-sample relationships across views; and ii) inability to adaptively utilize inter-view similarity, resulting in representation conflicts and semantic degradation. To address these issues, we propose a novel framework named Relationship Alignment for View-aware Multi-view Clustering (RAV). Our approach first constructs view-specific sample relationship matrices from deep features and aligns them with the global relationship matrix to enhance cross-view neighborhood consistency and facilitate accurate measurement of inter-view similarity. Simultaneously, we introduce a view-aware adaptive weighting mechanism for label contrastive learning that dynamically adjusts the contrastive intensity between view pairs based on deep-feature similarity: higher-similarity views lead to stronger label alignment, while lower-similarity views reduce the weighting to prevent enforcing agreement. This strategy promotes cluster-level semantic consistency while preserving natural inter-view relationships. Extensive experiments demonstrate that our method consistently outperforms state-of-the-art approaches on multiple benchmark datasets. Project website: https://github.com/chenzhe207/RAV.

## 1 Introduction

With the rapid growth of multi-modal data and big data, multi-view clustering (MVC) Chen et al. (2023b); Dong et al. (2023); Eisenberg et al. (2025); Trosten et al. (2023); Wan et al. (2024); Deng et al. (2024) has gained widespread attention in recent years. Compared to traditional single-view clustering, MVC can effectively integrate complementary information from different sources, capturing richer intrinsic data structures and leading to more accurate sample partitioning Liu et al. (2025); Wen et al. (2026). Consequently, MVC has been widely applied in numerous fields, including computer vision Xie et al. (2020), natural language processing Ke et al. (2024); Nadkarni et al. (2011), and social network analysis Fang et al. (2023b); Banez et al. (2022). Depending on the use of deep learning techniques, existing MVC methods can be broadly categorized into two types: traditional MVC methods and deep MVC methods.

The significant advancements in deep learning for representation learning have driven deep MVC Wen et al. (2023); Trosten et al. (2021); Xiao et al. (2025) to gradually become mainstream. Compared to traditional MVC, deep MVC leverages the powerful representation learning capabilities of deep neural networks and has demonstrated promising clustering performance. For instance, Lin et al. (2021) learns view-specific representations through intra-view reconstruction loss while maintaining cross-view consistency using mutual information-based contrastive learning. Xu et al. (2022) proposes a multi-level feature learning framework that alleviates the conflict between consistent representation learning and view-specific feature reconstruction. Yan et al. (2023) further employs the inter-sample structural relationships to guide contrastive learning, thereby achieving consistent representation

---

*Corresponding author.

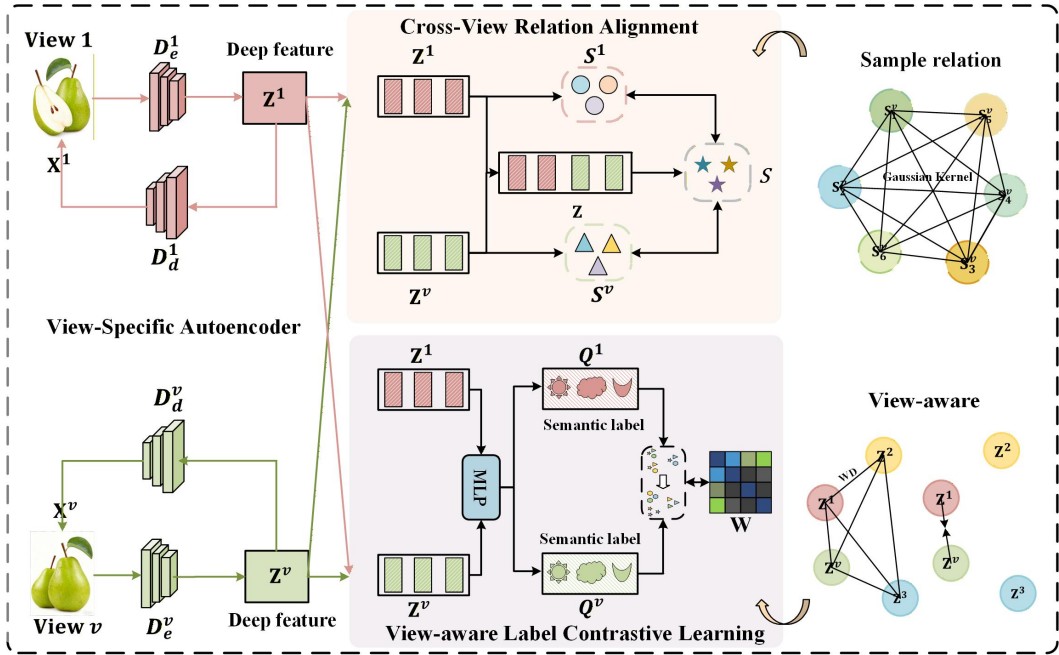

Figure 1: An illustration of the proposed RAV framework. The model crucially incorporates two modules: cross-view relation alignment to maintain neighborhood structures, and view-aware adaptive weighting in label contrastive learning to counteract representation degradation from view dissimilarity, thereby achieving robust multi-view clustering.

learning. These approaches effectively capture complex nonlinear relationships across views, thereby significantly improving both feature representation quality and clustering efficiency.

Meanwhile, contrastive learning Chen et al. (2025c); Dong et al. (2025); Chen et al. (2024; 2025b) can effectively capture high-level semantics while suppressing irrelevant information. Current contrastive learning-based deep MVC approaches primarily focus on two levels: sample-level and cluster-level. At the sample level, feature consistency is enhanced by bringing similar samples closer together and pushing dissimilar samples further apart. At the clustering level, the focus is on aligning the clustering distributions across views to achieve consistency in clustering semantics. For example, Li et al. (2021) introduces both sample-level and cluster-level contrastive objectives to jointly optimize feature representations and clustering assignments. Chen et al. (2023a) proposes cross-view cluster assignment contrastive learning to achieve consistent cluster distributions. Cui et al. (2024) develops a dual contrastive mechanism that systematically constructs positive and negative pairs for learning consistent representations. These research contributions provide novel insights for multi-view clustering tasks, while establishing a solid foundation for subsequent contrastive learning-based deep MVC methodologies.

Despite considerable advances in MVC, several challenges remain. Many methods fail to preserve the sample neighborhood structure between views, compromising the consistency of cross-view sample relationships. Moreover, most contrastive learning-based methods treat all views equally and force their alignment, which can disrupt semantic structures and cause representation conflicts when view differences are substantial. Although recent works Xu et al. (2023); Wu et al. (2024) attempt to mitigate representation degradation by introducing adaptive weighting at the feature level, some shortcomings remain. These methods primarily focus on introducing weight adjustment at the feature level. However, they fail to adequately consider the consistency of cross-view sample relational structures and the semantic consistency at the cluster level. Consequently, these approaches may still force low-similarity views to undergo consistency learning, leading to semantic conflicts and disruption of natural inter-view relationships. To overcome these issues, we propose a novel framework that combines sample relationship alignment with view-aware adaptive label contrastive learning. Our method first constructs a sample relationship matrix from deep features for each view and aligns it with the global relationship matrix. This effectively enhances cross-view neighborhood

consistency while preserving the local neighborhood structure of the samples. Meanwhile, cluster assignment matrices are generated via a shared MLP. To further enhance semantic consistency in clustering. We introduce a view-aware adaptive weighting strategy to adjust the contrastive strength based on the deep feature similarity between view pairs. Highly similar view pairs are assigned greater weight to enhance semantic consistency, while view pairs with low similarity are given smaller weight to avoid representation conflicts caused by forced alignment. This strategy effectively reduces representation degradation caused by view discrepancies and maintains the natural relationships between views.

The main contributions of this paper are summarized as follows:

- We introduce a global-guide-local sample relation alignment module that preserves neighborhood structures and enhances cross-view sample relationships consistency by aligning view-specific relation matrices with a global relation matrix.
- We propose a view-aware adaptive weighting mechanism for label contrastive learning, which dynamically adjusts the alignment strength based on the similarity between views, effectively mitigating semantic conflicts.
- Extensive experiments on multiple benchmarks show that our method achieves state-of-the-art performance across standard clustering metrics, demonstrating its effectiveness and generalizability.

## 2 RELATED WORK

### 2.1 DEEP MULTI-VIEW CLUSTERING

The advancement of deep learning has greatly propelled the development of deep MVC. Modern deep MVC methods capitalize on neural networks' strong nonlinear representation capabilities to effectively learn consistent, discriminative shared latent representations from multi-view data. These approaches can be broadly categorized into several lines of research. Graph-based methods focus on leveraging inter-sample relationships for representation learning and clustering. For instance, Fei et al. (2025) proposes a graph structure-aware contrastive clustering model that combines graph topology with node attribute features. Similarly, Wang et al. (2025) improves the quality and robustness of consensus graph learning through sample-level fusion and clustering effectiveness guidance. Subspace-based methods focus on deriving consistent representations across views. For example, Yu et al. (2025) introduces a pseudo-label-guided bidirectional discriminative subspace clustering framework that maintains structural coherence in sample affinity matrices. Lin et al. (2025) proposes a low-rank multi-view subspace clustering based on a Transformer autoencoder to learn a consistent and discriminative shared subspace representation. Reconstruction-based methods typically utilize autoencoders to learn a shared latent space and can reconstruct the original data or its representation. Yan et al. (2025), for instance, applies a self-supervised strategy to learn consistent view representations by reconstructing view features, achieving discriminative clustering under the constraint of view consistency.

### 2.2 CONTRASTIVE LEARNING

Contrastive learning, as a powerful unsupervised learning paradigm, demonstrates unique advantages in handling data heterogeneity and cross-view alignment tasks. It learns highly discriminative and consistent representations by constructing positive and negative sample pairs. Tang et al. (2020) proposes a decoupled contrastive MVC approach based on higher-order graph walks, achieving reliable clustering representations through intra-view and inter-view contrastive learning. Cui et al. (2024) designs a dual contrastive loss combining dynamic clustering diffusion and neighborhood-guided alignment, enhancing intra-cluster compactness while improving inter-cluster separation. Dong et al. (2025) further presents a view-graph-based progressive fusion method, achieving representation consistency through dual contrastive learning within and across views. Yi et al. (2025) attempts to leverage shared information to construct anchor points for contrastive learning, mitigating view-specificity and noise interference.

Despite existing methods having made significant progress, most remain limited to aligning features or clustering distributions, generally overlooking the construction of consistent sample relational

structures and failing to adequately address representation degradation caused by view discrepancies. In contrast, our approach maintains cross-view neighborhood consistency by aligning sample relational structures and adaptively leverages inter-view similarity to guide label contrastive learning. This effectively enhances semantic consistency without distorting natural view relationships.

# 3 METHOD

## 3.1 PRELIMINARIES

This section introduces our multi-view clustering framework, as depicted in Figure 1. The architecture consists of three core components: View-Specific Autoencoder Modules, a Cross-View Relation Alignment Module, and a View-aware Label Contrastive Learning Module. Given a multi-view dataset with $V$ views denoted as $\mathbf{X} = \{\mathbf{X}^1, \mathbf{X}^2, \ldots, \mathbf{X}^V\}$, where the data from the $v$-th view is represented as $\mathbf{X}^v = [\mathbf{x}_1^v; \mathbf{x}_2^v; \ldots; \mathbf{x}_N^v] \in \mathbb{R}^{N \times d_v}$, with $N$ being the number of samples and $d_v$ the feature dimension of view $v$, the framework is designed to jointly optimize these modules. This integrated approach effectively handles disparities in view similarity while promoting consistency in cross-view representations.

## 3.2 VIEW-SPECIFIC AUTOENCODER

In multi-view clustering, the quality of feature representations is critical to clustering performance. To extract robust latent features from raw multi-view data, which often contain noise and redundancy, we employ view-specific autoencoders for each view. Formally, for the $v$-th view, we define an encoder $f^v$ and a decoder $g^v$. The latent representation of the $i$-th sample in view $v$ is obtained as:

$$\mathbf{z}_{i,:}^v = f^v(\mathbf{x}_{i,:}^v; \theta^v), \tag{1}$$

where $\theta^v$ denotes the learnable parameters of the encoder for the $v$-th view, and $d$ is the dimensionality of the resulting latent feature. The latent representation $\mathbf{z}_i^v$ is then passed to the corresponding decoder $g^v$ to reconstruct the original input, formulated as:

$$\hat{\mathbf{x}}_{i,:}^v = g^v(\mathbf{z}_{i,:}^v; \phi^v) = g^v(f^v(\mathbf{x}_{i,:}^v; \theta^v); \phi^v), \tag{2}$$

where $\hat{\mathbf{x}}_{i,:}^v$ denotes the reconstruction of the $i$-th sample from the $v$-th view, and $\phi^v$ represents the trainable parameters of its decoder. The overall reconstruction loss for training all autoencoders is defined as:

$$\mathcal{L}_{REC} = \sum_{v=1}^{V} \sum_{i=1}^{N} \|\mathbf{x}_{i,:}^v - \hat{\mathbf{x}}_{i,:}^v\|_2^2. \tag{3}$$

## 3.3 CROSS-VIEW RELATIONSHIP ALIGNMENT

To enhance the consistency of sample relationships across views while preserving local neighborhood structures, we introduce a cross-view relation alignment module. First, deep features $\mathbf{Z}^v$ for each view are obtained using the view-specific encoders. The pairwise similarity between samples within a view is then computed via a Gaussian kernel, thereby constructing intra-view structural relationships. Specifically, the similarity between the $i$-th and $k$-th samples in the $v$-th view is given by:

$$s_{ik}^v = \exp\left(-\frac{\|\mathbf{z}_{i,:}^v - \mathbf{z}_{k,:}^v\|^2}{\sigma}\right), \tag{4}$$

where $s_{ik}^v$ denotes the similarity between the $i$-th and $k$-th samples in the $v$-th view. A smaller feature distance corresponds to higher similarity. Conversely, the larger the feature distance, the lower the similarity. Based on equation (4), we compute the pairwise similarities for each view and store them in a view-specific relation matrix $\{\mathbf{S}^v = [\mathbf{s}_1^v; \mathbf{s}_2^v; \ldots; \mathbf{s}_N^v] \in \mathbb{R}^{N \times N}\}_{v=1}^V$. To construct a global relation matrix that integrates information from all views, we first concatenate the deep features as follows:

$$\mathbf{Z} = \text{Concat}(\mathbf{Z}^1, \mathbf{Z}^2, \ldots, \mathbf{Z}^v), \tag{5}$$

where $\mathbf{Z} \in \mathbb{R}^{N \times (Vd)}$ represents the concatenated global features. The global relation matrix $\mathbf{S} = [\mathbf{s}_1; \mathbf{s}_2; \ldots; \mathbf{s}_N] \in \mathbb{R}^{N \times N}$ is then computed using the same similarity measure defined in equation (4). We align each view-specific relation matrix with this global matrix via a global-supervise-local contrastive learning objective, which pulls positive pairs closer while pushing negative pairs apart. The resulting cross-view relation alignment loss is formulated as:

$$\mathcal{L}_{\mathrm{S}} = -\frac{1}{N} \sum_{v=1}^{V} \sum_{i=1}^{N} \log \frac{e^{d(\mathbf{s}_{i,:}^{v}, \mathbf{s}_{i,:})/\tau_F}}{\sum_{k=1}^{N} e^{d(\mathbf{s}_{i,:}^{v}, \mathbf{s}_{k,:})/\tau_F} - e^{1/\tau_F}}, \tag{6}$$

where $\tau_F$ is a temperature hyperparameter, and $d(\mathbf{s}_{i,:}^{v}, \mathbf{s}_{k,:})$ denotes the cosine similarity function, defined as:

$$d(\mathbf{s}_{i,:}^{v}, \mathbf{s}_{k,:}) = \frac{\langle \mathbf{s}_{i,:}^{v}, \mathbf{s}_{k,:} \rangle}{\|\mathbf{s}_{i,:}^{v}\| \|\mathbf{s}_{k,:}\|}. \tag{7}$$

The alignment loss improves cross-view relational consistency by preserving sample neighborhood structures, ensuring that neighboring samples remain close while distant ones are separated. Consequently, it enhances both feature discriminability within views and semantic consistency across views.

## 3.4 VIEW-AWARE LABEL CONTRASTIVE LEARNING

After obtaining the view-specific representations, we project the features of each view through a shared MLP to generate the cluster assignment matrices $\{\mathbf{Q}^v \in \mathbb{R}^{N \times K}\}_{v=1}^{V}$. Each entry $q_{ij}^v$ denotes the probability that the $i$-th sample is assigned to the $j$-th cluster in the $v$-th view, obtained by applying the Softmax function along the cluster dimension. To promote clustering consistency across views, we apply contrastive learning at the cluster-assignment level. For the $j$-th cluster assignment vector $\mathbf{q}_{:,j}^v$ (i.e., the $j$-th column of $\mathbf{Q}^v$), we consider all possible assignment pairs across views and clusters, totaling $(VK - 1)$ pairs. Among these, pairs originating from the same cluster index $j$ but different views ($u \neq v$) are treated as positive pairs, amounting to $(V - 1)$ positives. The remaining $V(K - 1)$ pairs are considered negatives. The contrastive loss between view $v$ and view $u$ is defined as:

$$\ell_c^{(v,u)} = -\frac{1}{K} \sum_{j=1}^{K} \log \frac{e^{d(\mathbf{q}_{:,j}^{v}, \mathbf{q}_{:,j}^{u})/\tau_L}}{\sum_{k=1}^{K} \sum_{m=v,u} e^{d(\mathbf{q}_{:,j}^{v}, \mathbf{q}_{:,k}^{m})/\tau_L} - e^{1/\tau_L}}, \tag{8}$$

where $\tau_L$ is the temperature parameter of label contrastive learning. Afterward, the total label contrastive loss is given by:

$$\mathcal{L}_{\mathrm{Q}} = \frac{1}{2} \sum_{v=1}^{V} \sum_{u \neq v} \ell_c^{(v,u)} + \sum_{v=1}^{V} \sum_{j=1}^{K} r_j^v \log r_j^v, \tag{9}$$

where $r_j^v = \frac{1}{N} \sum_{i=1}^{N} q_{ij}^v$. The first term enhances cross-view clustering consistency, while the second acts as a regularization term that prevents all samples from being assigned to a single cluster, thereby avoiding trivial solutions.

However, this approach does not account for inherent feature distribution discrepancies across views. When contrastive learning forcibly aligns view pairs with substantial differences, it may distort genuine semantic structures and cause representation degradation. To address this issue, we propose a view-aware adaptive weighting strategy for label contrastive learning, which dynamically modulates the influence of each view pair based on their deep feature similarity. Specifically, we first employ the Wasserstein Distance (WD) Shen et al. (2018) to quantify the discrepancy between the feature distributions of two views. The WD between view $v$ and view $u$ is defined as:

$$WD(\mathbf{Z}^v, \mathbf{Z}^u) = \frac{1}{N^2} \sum_{i=1}^{N} \sum_{k=1}^{N} |\mathbf{z}_{i,:}^{v} - \mathbf{z}_{k,:}^{u}|, \tag{10}$$

where $\mathbf{z}_{i,:}^{v}$ and $\mathbf{z}_{k,:}^{u}$ denote the deep features of the $i$-th sample in view $v$ and the $k$-th sample in view $u$, respectively. The adaptive weight between views $v$ and $u$ is then calculated based on their

Wasserstein Distance as follows:

$$w_{(v,u)} = \frac{e^{-WD(\mathbf{Z}^v, \mathbf{Z}^u)}}{\sum\limits_{u=1}^{V} e^{-WD(\mathbf{Z}^v, \mathbf{Z}^u)}}, \tag{11}$$

where $w_{(v,u)}$ denotes the adaptive weight between the representations $\mathbf{Z}^v$ and $\mathbf{Z}^u$. The pairwise weights of all views form a $V \times V$ matrix $\mathbf{W}$. This matrix enables a dynamic weighting strategy: view pairs with high feature similarity (i.e., small WD values) are assigned larger weights to strengthen their contribution during contrastive learning, while pairs with large representation discrepancies (i.e., high WD values) are assigned smaller weights to reduce potential negative effects. Integrating this weighting mechanism into the label contrastive learning objective, we obtain the final view-aware adaptive weighting loss:

$$\mathcal{L}_{\mathrm{Q}} = \frac{1}{2}\sum_{v=1}^{V}\sum_{u \neq v}\frac{1}{2}(w_{(v,u)} + w_{(u,v)})\ell_c^{(v,u)} + \sum_{v=1}^{V}\sum_{j=1}^{K} r_j^v \log r_j^v. \tag{12}$$

### 3.5 The Overall Loss Function

Based on the foregoing formulation, the overall objective function integrates the three key components as follows:

$$\mathcal{L}_{\mathrm{total}} = \mathcal{L}_{\mathrm{REC}} + \lambda_1 \mathcal{L}_{\mathrm{Q}} + \lambda_2 \mathcal{L}_{\mathrm{S}}, \tag{13}$$

where $\lambda_1$ and $\lambda_2$ are balancing coefficients, $\mathcal{L}_{\mathrm{REC}}$ denotes the reconstruction loss, $\mathcal{L}_{\mathrm{Q}}$ represents the view-aware adaptive label contrastive loss, and $\mathcal{L}_{\mathrm{S}}$ corresponds to the cross-view relation alignment loss.

Once the model converges, the clustering labels can be obtained as follows:

$$y_j = \arg\max_{j}\left(\frac{1}{V}\sum_{v=1}^{V} q_{ij}^v\right). \tag{14}$$

The full process of our RAV is summarized in Algorithm 1.

---

**Algorithm 1** : The optimization of RAV.

---

1: **Input**: Multi-view dataset $\{\mathbf{X}^v\}_{v=1}^{V}$; The number of samples is $N$; The number of max epochs is $T$; The number of clusters is $K$; The parameters $\lambda_1, \lambda_2$.
2: **Initialization:** Initialize autoencoder parameters by minimizing $\mathcal{L}_{\mathrm{REC}}$ in Eq. (3).
3: (using mini-batch training)
4: **for** $t = 1$ to T **do**
5:     Obtain the relationship matrix $\mathbf{S}^v$ for the specific view and the global relationship matrix $\mathbf{S}$ by Eq. (4).
6:     Obtain the weight matrix $\mathbf{W}$ by Eq. (11).
7:     Optimize $\{\theta^v, \phi^v\}_{v=1}^{V}$ by minimizing $\mathcal{L}_{\mathrm{total}}$ in Eq. (13) .
8: **end for**
9: Calculate the predicted labels by Eq. (14).
10: **Output**: $\mathbf{Y} = [y_1, y_2, \ldots, y_N]$.

---

## 4 Experiment

### 4.1 Datasets and Experimental Setting

**Datasets.** We evaluate our model on ten benchmark datasets. Table 1 summarizes their key characteristics, including sample size, number of views, number of clusters, and feature dimensions for each view. NGs Yan et al. (2025): This dataset consists of 500 documents, which have been preprocessed using three different methods to obtain three distinct views. Digit-Product Cui et al. (2024): This

dataset is derived from MNIST and Fashion Handwritten digits, containing 30,000 samples and two views. ALOI Cui et al. (2024): This dataset contains 10,800 samples and 10 clusters, with four views extracted from each image, representing color similarity, Haralick, HSV, and RGB features. Cora Fang et al. (2023a): This dataset contains 2,708 documents, with four features selected as the four views: content, inbound, outbound, and citations. It is categorized into seven clusters. NUSWIDE Chua et al. (2009): This dataset consists of 5,000 images, classified into 5 categories. Caltech-5V Xu et al. (2022): This dataset is an RGB image dataset containing 1,400 images, covering WM, CENTRIST, LBP, GIST, and HOG features. NoisyMNIST Wang et al. (2015): This dataset comprises 50,000 samples, organized into 10 clusters. YoutubeVideo Madani et al. (2012): This dataset consists of 101,499 samples, divided into 31 classes. 3Sources[1]: This dataset contains 169 samples, 3 views, and 6 classes. Fashion Xiao et al. (2017): This dataset contains 10,000 images, 3 views, and 10 clusters.

Table 1: Description of the used multi-view datasets.

| Dataset | Samples | Views | Clusters | Dimensionality |
|---|---|---|---|---|
| NGs | 500 | 3 | 5 | 2000/2000/2000 |
| Digit-Product | 30,000 | 2 | 10 | 1024/1024 |
| ALOI | 10,800 | 4 | 100 | 77/13/64/125 |
| Cora | 2,708 | 4 | 7 | 2708/1433/2708/2708 |
| NUSWIDE | 5,000 | 5 | 5 | 65/226/145/74/129 |
| Caltech-5V | 1,400 | 5 | 7 | 40/254/928/512/1984 |
| NoisyMNIST | 50,000 | 2 | 10 | 784/784 |
| YoutubeVideo | 101,499 | 3 | 31 | 512/647/838 |
| 3Sources | 169 | 3 | 6 | 3560/3631/3068 |
| Fashion | 10,000 | 3 | 10 | 784/784/784 |

**Implementation Details.** All experiments are implemented in PyTorch 1.12.1 on an NVIDIA RTX 4090 D GPU. The model is optimized using Adam with a fixed learning rate of 0.0003 and a batch size of 256. Both pretraining and fine-tuning phases are set to 200 epochs. We introduce two hyperparameters $\lambda_1$ and $\lambda_2$, with ranges of $[0.00001, 0.0001, \ldots, 1000]$ and $[0.00001, 0.0001, \ldots, 1]$, respectively. The Gaussian kernel bandwidth $\sigma$ is set to 1.0, and both $\tau_F$ and $\tau_L$ are fixed to 0.5. All baseline methods are conducted in the same experimental setup, using either the parameters recommended in the baseline methods or those optimized through parameter search for the best performance.

## 4.2 COMPARED METHODS AND RESULTS

We compare nine representative multi-view clustering methods across ten benchmark datasets to evaluate our approach. MFLVC Xu et al. (2022): This method is primarily used for multi-level feature learning in multi-view clustering. GCFAgg Yan et al. (2023): This method mainly utilizes sample similarity structures to guide contrastive learning. SEM Xu et al. (2023): This method mostly guides contrastive learning by adjusting view weights. MVCAN Xu et al. (2024): This method alleviates the negative impact of noisy views and optimizes the learning of individual image representations. SCMVC Wu et al. (2024): This method adaptively strengthens useful views in feature fusion while weakening unreliable views. DDMVC Xu et al. (2025): This method considers diversity and discriminative feature learning. SSLNMVC Yan et al. (2025): This method introduces the UProjection module, which enhances the expressiveness of consistent features by feature resampling and concatenating the fused features before and after resampling. AICN-MLM Shu et al. (2025): This method proposes a fuzzy instance-aware multi-level matching contrastive network for multi-view document clustering. DFL-NET Chen et al. (2025a): This method utilizes orthogonal constraints to separate shared and unique features, then ensures consistency in cross-view clustering through cross-view label comparisons.

We evaluate our method using three widely recognized clustering evaluation metrics: Accuracy (ACC), Normalized Mutual Information (NMI), and Purity (PUR), as shown in Tables 2, 3, and 4. Based on these results, we can draw the following conclusions:

---

[1] http://mlg.ucd.ie/datasets/3sources.html

Table 2: Clustering results of all methods on the NGs, Digit-Product, and ALOI datasets.

| Datasets | NGs | | | Digit-Product | | | ALOI | | |
|---|---|---|---|---|---|---|---|---|---|
| Evaluation Metrics | ACC | NMI | PUR | ACC | NMI | PUR | ACC | NMI | PUR |
| MFLVC (22 CVPR) | 0.932 | 0.825 | 0.932 | 0.991 | 0.976 | 0.991 | 0.435 | 0.786 | 0.435 |
| GCFAgg (23 CVPR) | 0.894 | 0.742 | 0.894 | 0.988 | 0.968 | 0.988 | 0.790 | 0.917 | 0.809 |
| SEM (24 NeurIPS) | 0.856 | 0.673 | 0.856 | 0.991 | 0.976 | 0.991 | 0.771 | 0.899 | 0.787 |
| MVCAN (24 CVPR) | 0.470 | 0.271 | 0.470 | 0.989 | 0.967 | 0.989 | **0.849** | **0.929** | **0.864** |
| SCMVC (24 TMM) | 0.826 | 0.638 | 0.826 | 0.995 | 0.986 | 0.995 | 0.818 | 0.925 | 0.835 |
| DDMVC (25 PR) | – | – | – | 0.968 | 0.931 | 0.968 | 0.796 | 0.907 | 0.813 |
| SSLNMVC (25 TMM) | 0.936 | 0.842 | 0.936 | 0.990 | 0.973 | 0.990 | 0.541 | 0.814 | 0.558 |
| AICN-MLM (25 AAAI) | 0.912 | 0.774 | 0.912 | 0.991 | 0.976 | 0.991 | 0.788 | 0.906 | 0.800 |
| DFL-NET (25 TKDE) | 0.904 | 0.786 | 0.904 | **0.998** | **0.993** | **0.998** | 0.825 | 0.915 | 0.832 |
| ours | **0.980** | **0.934** | **0.980** | **0.998** | **0.993** | **0.998** | 0.826 | 0.912 | 0.830 |

Table 3: Clustering results of all methods on the Cora, NUSWIDE, and Caltech-5V datasets.

| Datasets | Cora | | | NUSWIDE | | | Caltech-5V | | |
|---|---|---|---|---|---|---|---|---|---|
| Evaluation metrics | ACC | NMI | PUR | ACC | NMI | PUR | ACC | NMI | PUR |
| MFLVC (22 CVPR) | 0.268 | 0.111 | 0.377 | 0.624 | 0.338 | 0.624 | 0.867 | 0.781 | 0.867 |
| GCFAgg (23 CVPR) | 0.220 | 0.051 | 0.304 | 0.596 | 0.336 | 0.596 | 0.799 | 0.697 | 0.799 |
| SEM (24 NeurIPS) | 0.220 | 0.028 | 0.313 | 0.588 | 0.317 | 0.588 | 0.901 | 0.834 | 0.901 |
| MVCAN (24 CVPR) | 0.567 | 0.385 | **0.640** | 0.572 | 0.290 | 0.572 | **0.919** | **0.856** | **0.919** |
| SCMVC (24 TMM) | 0.311 | 0.160 | 0.388 | 0.622 | 0.354 | 0.622 | 0.871 | 0.783 | 0.871 |
| DDMVC (25 PR) | 0.323 | 0.141 | 0.407 | 0.607 | 0.312 | 0.636 | 0.771 | 0.695 | 0.779 |
| SSLNMVC (25 TMM) | 0.277 | 0.102 | 0.364 | 0.637 | 0.367 | 0.637 | 0.881 | 0.789 | 0.881 |
| AICN-MLM (25 AAAI) | 0.331 | 0.171 | 0.418 | 0.612 | 0.337 | 0.612 | 0.898 | 0.828 | 0.898 |
| DFL-NET (25 TKDE) | 0.359 | 0.185 | 0.435 | 0.614 | 0.325 | 0.614 | 0.833 | 0.767 | 0.833 |
| ours | **0.592** | **0.404** | 0.598 | **0.647** | **0.371** | **0.647** | 0.901 | 0.839 | 0.901 |

Table 4: Clustering results of all methods on the NoisyMNIST, YoutubeVideo, 3Sources, and Fashion datasets.

| Datasets | NoisyMNIST | | | YoutubeVideo | | | 3Sources | | | Fashion | | |
|---|---|---|---|---|---|---|---|---|---|---|---|---|
| Evaluation metrics | ACC | NMI | PUR | ACC | NMI | PUR | ACC | NMI | PUR | ACC | NMI | PUR |
| MFLVC (22 CVPR) | 0.988 | 0.965 | 0.988 | 0.238 | 0.224 | 0.324 | 0.521 | 0.477 | 0.669 | **0.994** | **0.985** | **0.994** |
| GCFAgg (23 CVPR) | 0.781 | 0.847 | 0.836 | 0.275 | 0.263 | 0.361 | 0.521 | 0.429 | 0.615 | 0.990 | 0.974 | 0.990 |
| SEM (24 NeurIPS) | 0.995 | 0.984 | 0.995 | 0.318 | 0.309 | 0.404 | 0.533 | 0.584 | 0.716 | **0.994** | 0.983 | **0.994** |
| MVCAN (24 CVPR) | 0.933 | 0.861 | 0.933 | 0.244 | 0.244 | 0.341 | 0.562 | 0.478 | 0.663 | 0.856 | 0.840 | 0.856 |
| SCMVC (24 TMM) | **0.996** | **0.988** | **0.996** | 0.291 | 0.277 | 0.376 | 0.562 | 0.420 | 0.627 | **0.994** | **0.985** | **0.994** |
| DDMVC (25 PR) | 0.957 | 0.893 | 0.957 | – | – | – | 0.456 | 0.346 | 0.592 | 0.931 | 0.892 | 0.931 |
| SSLNMVC (25 TMM) | 0.995 | 0.985 | 0.995 | 0.235 | 0.244 | 0.430 | 0.521 | 0.510 | 0.686 | **0.994** | 0.984 | **0.994** |
| AICN-MLM (25 AAAI) | 0.990 | 0.971 | 0.990 | – | – | – | 0.538 | 0.472 | 0.675 | **0.994** | **0.985** | **0.994** |
| DFL-NET (25 TKDE) | 0.995 | 0.983 | 0.995 | 0.227 | 0.246 | 0.6 | **0.586** | 0.572 | 0.728 | 0.993 | 0.982 | 0.993 |
| ours | **0.996** | 0.986 | **0.996** | **0.356** | **0.332** | **0.445** | 0.574 | **0.599** | **0.775** | **0.994** | 0.984 | **0.994** |

Our method outperforms baseline approaches on most datasets, confirming its effectiveness. Notably, on the NGs, YoutubeVideo, and Cora datasets, the ACC metric improves by 4.4%, 7.8%, and 2.5%, respectively, compared to the second-best results. These gains stem from the proposed relation alignment and view-aware adaptive label contrastive learning mechanism. The relational alignment module maintains structural consistency between views, providing a stable structure for view similarity measurement and contrastive learning. Meanwhile, the view-aware adaptive weighting mechanism effectively mitigates representation conflicts caused by view differences, preventing the forced alignment of low-similarity views and thereby improving clustering accuracy.

On ALOI and Caltech-5V, our method performs slightly below MVCAN. This may be because MVCAN does not adopt standard contrastive learning, which reduces its sensitivity to view differences. On the Fashion dataset, our results are comparable to MFLVC, AICN-MLM, and SSLNMV. This may be attributed to the relatively simple structure of the dataset and the small view differences, thereby

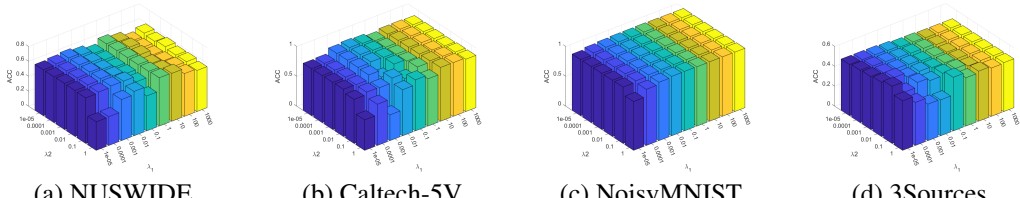

(a) NUSWIDE        (b) Caltech-5V        (c) NoisyMNIST        (d) 3Sources

Figure 2: Parameter sensitivity analysis of $\lambda_1$ and $\lambda_2$ on the NUSWIDE, Caltech-5V, NoisyMNIST, and 3Sources datasets.

reducing the necessity of the adaptive weighting mechanism. On the Digit-Product dataset, RAV performs similarly to DFL-NET, which mitigates the impact of view inconsistency to some extent by decoupling shared representations from view-specific representations. However, RAV outperforms these methods on most other more challenging datasets, further validating the effectiveness of the adaptive weighting mechanism.

Compared to the feature-level contrastive weighting-based SEM and SCMVC methods, our approach achieves comparable performance on simpler datasets like NoisyMNIST and Fashion, while significantly outperforming them on more complex datasets. This demonstrates that the WD based on deep features can more accurately capture the intrinsic similarity between views. Therefore, when guiding label-based contrastive learning, it exhibits superior generalization capability and better preserves natural inter-view relationships.

## 4.3 MODEL ANALYSES

**Parameter Sensitivity:** To evaluate the impact of $\lambda_1$ and $\lambda_2$ on the model, we conduct a parameter sensitivity analysis as illustrated in Figure 2. The experiments set $\lambda_1$ within the range $[10^{-5}, 10^{-4}, 10^{-3}, 10^{-2}, 10^{-1}, 1, 10, 10^2, 10^3]$ and $\lambda_2$ within the range $[10^{-5}, 10^{-4}, 10^{-3}, 10^{-2}, 10^{-1}, 1]$. The results demonstrate that when these hyperparameters vary within the specified ranges, the clustering performance on all four datasets exhibits only minor fluctuations. This relatively small variation in performance proves that our method is highly robust to hyperparameter selection.

**Convergence:** We observe the changes in training loss and evaluation metrics (ACC/NMI) across four benchmark datasets to analyze the convergence characteristics of the proposed method. Figure 3 shows the convergence curves for the Caltech-5V, Digit-Product, NGs, and NoisyMNIST datasets. The main observations are as follows: First, the loss function decreases rapidly in the initial training phase and then gradually stabilizes until convergence. Second, the clustering evaluation metrics ACC and NMI continuously increase during training and eventually stabilize. Finally, this convergence trend indicates that our loss function effectively regularizes the model and drives parameter optimization. These results not only demonstrate the convergence stability of our method but also validate its effectiveness in improving clustering performance.

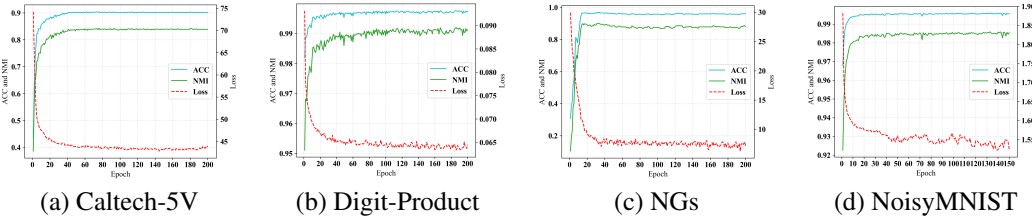

(a) Caltech-5V        (b) Digit-Product        (c) NGs        (d) NoisyMNIST

Figure 3: Convergence analysis of ours on Caltech-5V, Digit-Product, NGs, and NoisyMNIST datasets, where each subgraph displays the total loss and both evaluation metrics (ACC/NMI) over training epochs.

**Visualization Analysis:** To validate our method's ability to obtain a user-friendly clustering structure, we visualize the learned global feature $\mathbf{Z}$ on the Digit-Product dataset using the t-SNE Maaten & Hinton (2008) method. As shown in the Figure 4, with increasing iterations of training, the clustering structure of $\mathbf{Z}$ becomes increasingly distinct. The distance between different clusters grows larger, while clusters within the same group become increasingly compact.

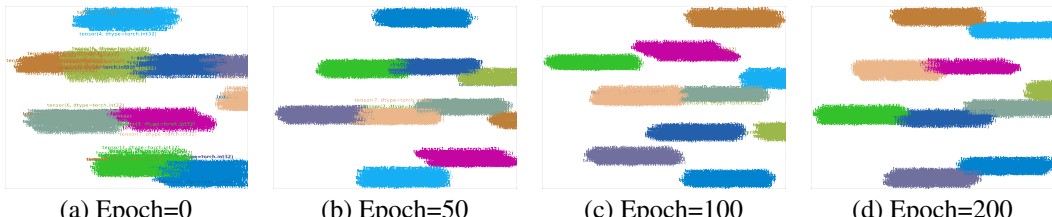

|  (a) Epoch=0 | (b) Epoch=50 | (c) Epoch=100 | (d) Epoch=200 |

Figure 4: Visualization of the global feature $\mathbf{Z}$ for cascading on Digit-Product dataset. The same color indicates features belonging to the same cluster.

Table 5: Ablation studies on different loss components on the Caltech-5V, NUSWIDE, ALOI, and 3Sources datasets.

| Components | | | Caltech-5V | | | NUSWIDE | | | ALOI | | | 3Sources | | |
|---|---|---|---|---|---|---|---|---|---|---|---|---|---|---|
| $\mathcal{L}_{REC}$ | $\mathcal{L}_Q$ | $\mathcal{L}_S$ | ACC | NMI | PUR | ACC | NMI | PUR | ACC | NMI | PUR | ACC | NMI | PUR |
| ✓ | ✓ | ✗ | 0.899 | 0.830 | 0.899 | 0.644 | 0.362 | 0.644 | 0.780 | 0.887 | 0.789 | 0.562 | 0.464 | 0.686 |
| ✓ | ✗ | ✓ | 0.424 | 0.309 | 0.439 | 0.298 | 0.037 | 0.311 | 0.264 | 0.656 | 0.264 | 0.396 | 0.135 | 0.408 |
| ✓ | ✓ | ✓ | **0.901** | **0.839** | **0.901** | **0.647** | **0.371** | **0.647** | **0.826** | **0.912** | **0.830** | **0.574** | **0.599** | **0.775** |

Table 6: Ablation study on the view-aware adaptive weighting mechanism for NGs, Digit-Product, ALOI and Cora datasets.

| Datasets | NGs | | | Digit-Product | | | ALOI | | | Cora | | |
|---|---|---|---|---|---|---|---|---|---|---|---|---|
| Evaluation metrics | ACC | NMI | PUR | ACC | NMI | PUR | ACC | NMI | PUR | ACC | NMI | PUR |
| ours w/o W | 0.966 | 0.895 | 0.966 | **0.998** | **0.993** | **0.998** | 0.801 | 0.903 | 0.807 | 0.585 | 0.393 | 0.585 |
| ours | **0.980** | **0.934** | **0.980** | **0.998** | **0.993** | **0.998** | **0.826** | **0.912** | **0.830** | **0.592** | **0.404** | **0.598** |

**Ablation Studies:** To systematically evaluate the contribution of each component, we conduct a comprehensive ablation study. As summarized in Table 5, $\mathcal{L}_{REC}$, $\mathcal{L}_Q$, and $\mathcal{L}_S$ represent the reconstruction loss, the view-aware adaptive weighting contrastive loss, and the cross-view relation alignment loss, respectively. Experiments for four benchmark datasets—Caltech-5V, NUSWIDE, ALOI, and 3Sources—show that the full model outperforms variants that remove either the relation alignment module or the view-aware weighting mechanism. These results confirm that relation alignment helps capture semantic structures across views, while adaptive weighting enhances robustness by dynamically moderating the influence of view pairs.

We further examine the specific contribution of the view-aware weighting strategy in Table 6. On the NGs, ALOI, and Cora datasets, the full model achieves ACC gains of 1.4%, 3.6%, and 0.7%, respectively, compared to the variant without weighting (Ours w/o W). These improvements indicate that the weighting strategy effectively alleviates representation degradation resulting from view discrepancy. On the Digit-Product dataset, however, performance remains unchanged—likely due to its inherently small inter-view differences, which diminish the need for adaptive weighting. This outcome underscores the particular usefulness of our weighting mechanism in scenarios with pronounced view disparities.

## 5 CONCLUSION

This paper presents a multi-view clustering framework that integrates sample relation alignment with view-aware adaptive weighting for contrastive learning. The framework employs a globally guided relational alignment module to enhance the consistency of neighborhood structures across multiple views and introduces a view-aware adaptive weighting mechanism based on WD to mitigate the negative effects of view similarity discrepancies. The stable neighborhood structure provided by the relationship alignment facilitates more accurate measurement of inter-view differences, enabling more reliable semantic alignment through view-aware adaptive weighting contrastive learning. Extensive experiments show that the proposed method effectively mitigates the impact of view disparity and outperforms existing approaches on multiple benchmarks. Future work will focus on theoretically exploring more robust relational structures and universal similarity measures, while practically extending the method to complex scenarios such as incomplete views and noisy data to further improve its capability in handling view discrepancy challenges.

## 6 ACKNOWLEDGMENT

This work was supported in part by the National Natural Science Foundation of China (62306006, 62576152, 62332008, 62336004), the Basic Research Program of Jiangsu (BK20250104), the Fundamental Research Funds for the Central Universities (JUSRP202504007).

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
