# OpenReview forum: "Relationship Alignment for View-aware Multi-view Clustering"
_ICLR.cc/2026/Conference — ICLR 2026 Poster_

### Official Review · Reviewer_R3ke · 2025-10-23

**Soundness:** 4
**Presentation:** 3
**Contribution:** 4
**Rating:** 8
**Confidence:** 5

**Summary:**

The approach is well-motivated, combining relation alignment with view similarity measurement to offer a novel perspective for addressing semantic degradation caused by view discrepancies in multi-view clustering.

**Strengths:**

1. This paper effectively addresses two core issues in current deep multi-view clustering: the weak preservation of neighborhood structure and the representation degeneration caused by view discrepancies. It directly offers solutions to these problems by introducing two modules: cross-view relationship alignment and view-aware label contrastive learning. The former ensures the consistency of the neighborhood structure through alignment, while the latter dynamically adjusts the inter-view weights based on the Wasserstein distance. The relational alignment module and the view-aware weighting mechanism are ingeniously designed, and their combination exhibits strong synergistic effects.
2. The experiments cover multiple datasets, involving different types, sample sizes, and numbers of views, fully showcasing the diversity of the data.
3. The experiments include sensitivity analysis of parameters, convergence analysis, and ablation studies, validating the effectiveness of the proposed method from multiple perspectives.

**Weaknesses:**

1. This paper has some shortcomings in terms of literature coverage in the relevant field, as it fails to comprehensively showcase the depth of the research.
2. The description of the experimental details is insufficient, such as the lack of clarification on the choice of the Gaussian kernel bandwidth σ.
3. The description of the formulas is somewhat confusing, particularly in the case of Equation (6).

**Questions:**

1. In terms of references, it is recommended that the author include citations to more relevant literature, especially recent studies in the field, to enhance the completeness and reference value of the paper.
2. Although the experimental data is already comprehensive, it is recommended to analyze in more depth the performance of the method on different datasets, particularly the reasons why it performs better or worse on certain datasets.
3. Although the paper mentions the differences between this method and the SEM method, the explanation is still not clear enough. Please further elaborate on the core differences between the two.
4. This paper primarily uses the Wasserstein Distance (WD) to quantify the differences between views. It would be useful to explain why the WD was chosen as the metric, rather than other common metrics like MMD or cosine similarity.
5. In the calculation of the relationship matrix, the paper does not specify the value of the Gaussian kernel bandwidth σ, nor does it indicate whether σ was adaptively adjusted for different datasets.
6. In Equation (6), the parameters τ_F and τ_L are mentioned, but their specific meanings are not clearly defined. If the lack of clarity is due to a presentation issue or if they carry special significance, further explanation is needed to enhance the clarity and accuracy of the equation.

---

> ### Author Response · Authors · 2025-11-27
>
> **Response to weakness 1 and question 1:**
>
> We appreciate the reviewers' constructive feedback. We have expanded the Introduction and Related Work sections as suggested. During this expansion, we incorporated representative literature and recent advancements in the field of deep multi-view clustering over the past few years, thereby providing a more comprehensive and cutting-edge background overview of this research domain. Additionally, we have refined our analysis of existing method limitations and standardized the formulas and notation throughout the paper. Through these revisions, we aim to strengthen the paper's theoretical foundation, enhance the clarity of our research motivation, and provide more coherent and persuasive theoretical support for the proposed method. We believe these improvements will enhance the paper's overall structure and logical rigor while better showcasing the unique value and contributions of our research.
>
> **Response to weakness 2 and question 5:**
>
> We appreciate the reviewers' valuable comments. Regarding the Gaussian kernel bandwidth parameter $\sigma$ used in the relationship matrix calculation, it controls the scale at which similarity between samples is calculated. A larger σ value causes samples to exhibit higher similarity with more samples. A smaller $\sigma$ value causes samples to exhibit high similarity with those that are very close and highly similar. In our experiments, we uniformly set $\sigma$ to 1.0 without adaptive adjustments for different datasets. This approach primarily simplifies model complexity and ensures consistency across diverse scenarios. Despite the fixed $\sigma$, our method demonstrates stable and outstanding performance across multiple datasets. Settings for other relevant parameters and experimental details are also explicitly outlined in the Implementation Details section. We believe that supplementing these details effectively enhances the reproducibility of the experiments and further improves the completeness of the method proposed in this paper.
>
> **Response to weakness 3 and question 6:**
>
> We appreciate the valuable feedback provided by the reviewers. Regarding the ambiguity in Equation (6), we acknowledge this was due to a typographical error. $τ_F$ and $τ_L$ are temperature parameters. $τ_F$ is used to regulate the distribution of similarity between sample relationships in the relationship alignment loss, while $τ_L$ controls the distribution of similarity in the clustering assignment within the label contrast loss. Due to our oversight, this distinction was not accurately and clearly expressed in the formula. Following your feedback, we have thoroughly reviewed and revised all symbols and equations in the paper to ensure clear and consistent definitions for each notation. Equation (6) has also been rewritten to accurately reflect its meaning. We sincerely appreciate the reviewers' assistance in identifying and refining this critical detail. We believe these revisions will significantly enhance the paper's accuracy and readability.
>
> $L_{{S}} = -\frac{1}{N} \sum_{v=1}^V \sum_{i=1}^N \log \frac{e^{d(s_{i,:}^v, s_{i,:})/\tau_F}}{\sum_{k=1}^N e^{d(s_{i,:}^v, s_{k,:})/\tau_F} - e^{1/\tau_F}}$

---

> ### Author Response · Authors · 2025-11-27
>
> **Response to question 2:**
>
> We sincerely appreciate your constructive suggestions. We have conducted a more in-depth summary and analysis of the experimental results, particularly supplementing comparisons with two relevant methods SCMVC [1] and DFL-NET [2] to comprehensively evaluate the effectiveness of our approach. Through detailed analysis of performance across different datasets, we highlight the applicability and innovation of our method. Furthermore, the newly added comparisons with SCMVC and DFL-NET demonstrate that our approach possesses unique advantages. Particularly when handling scenes with significant view variations or complex structures, our method exhibits enhanced robustness.
>
> Table2: Clustering results of all methods on the NGs, Digit-Product, and ALOI datasets.
>
> | Datasets           |             NGs              |           Digit-Product          |              ALOI             |
> |--------------------|:----------------:|:----------------:|:----------------:|
> | Evaluation Metrics | ACC  NMI  PUR    | ACC  NMI  PUR    | ACC  NMI  PUR    |
> | MFLVC (22 CVPR)    | 0.932  0.825  0.932 | 0.991  0.976  0.991 | 0.435  0.786  0.435 |
> | GCFAgg (23 CVPR)   | 0.894  0.742  0.894 | 0.988  0.968  0.988 | 0.790  0.917  0.809 |
> | SEM (24 NeurIPS)   | 0.856  0.673  0.856 | 0.991  0.976  0.991 | 0.771  0.899  0.787 |
> | MVCAN (24 CVPR)    | 0.470  0.271  0.470 | 0.989  0.967  0.989 | **0.849**  **0.929**  **0.864** |
> | SCMVC (24 TMM)     | 0.826  0.638  0.826 | *0.995*  *0.986*  *0.995* | 0.818  *0.925*  *0.835* |
> | DDMVC (25 PR)      | --         --      --| 0.968  0.931  0.968 |0.796  0.907 0.813 |
> | SSLNMVC (25 TMM)   | *0.936*  *0.842*  *0.936* | 0.990  0.973  0.990 | 0.541  0.814  0.558 |
> | AICN-MLM (25 AAAI) | 0.912  0.774  0.912 | 0.991  0.976  0.991 | 0.788  0.906  0.800 |
> | DFL-NET (25 TKDE)  | 0.904  0.786  0.904 | **0.998**  **0.993**  **0.998** | 0.825  0.915  0.832 |
> | ours               | **0.980**  **0.934**  **0.980** | **0.998**  **0.993**  **0.998** | *0.826*  0.912  *0.830* |
>
> Table 3: Clustering results of all methods on the Cora, NUSWIDE, and Caltech-5V datasets.
>
> | Datasets      | Cora                 | NUSWIDE             | Caltech-5V          |
> | :------------ | :------------------- | :------------------ | :------------------ |
> | Evaluation Metrics  | ACC NMI PUR      | ACC NMI PUR    | ACC NMI PUR     |
> | MFLVC (22 CVPR)     | 0.268 0.111 0.377   | 0.624 0.338 0.624   | 0.867 0.781 0.867   |
> | GCFAgg (23 CVPR)    | 0.220 0.051 0.304   | 0.596 0.336 0.596   | 0.799 0.697 0.799   |
> | SEM (24 NeurIPS)    | 0.220 0.028 0.313   | 0.588 0.317 0.588   | *0.901* 0.834 *0.901*   |
> | MVCAN (24 CVPR)     | *0.567* *0.385* **0.640** | 0.572 0.290 0.572   | **0.919** **0.856** **0.919** |
> | SCMVC (24 TMM)      | 0.311 0.160 0.388   | 0.622 0.354 0.622   | 0.871 0.783 0.871   |
> | DDMVC (25 PR)       | 0.323 0.141 0.407   | 0.607 0.312 0.636   | 0.771 0.695 0.779   |
> | SSLNMVC (25 TMM)    | 0.277 0.102 0.364   | *0.637* *0.367* *0.637*   | 0.881 0.789 0.881   |
> | AICN-MLM (25 AAAI)  | 0.331 0.171 0.418   | 0.612 0.337 0.612   | 0.898 0.828 0.898   |
> | DFL-NET (25 TKDE)   | 0.359 0.185 0.435   | 0.614 0.325 0.614   | 0.833 0.767 0.833   |
> | ours            | **0.592** **0.404** *0.598* | **0.647** **0.371** **0.647** | *0.901* *0.839* *0.901*   |
>
> Table 4: Clustering results of all methods on the NoisyMNIST, YoutubeVideo, 3Sources, and Fashion datasets.
>
> | Datasets      | NoisyMNIST          | YoutubeVideo        | 3Sources            | Fashion             |
> | :------------ | :------------------ | :------------------: | :-----------------: | :------------------: |
> | Evaluation Metrics   | ACC NMI PUR    | ACC NMI PUR  | ACC NMI PUR   | ACC NMI PUR   |
> | MFLVC (22 CVPR)     | 0.988 0.965 0.988   | 0.238 0.224 0.324   | 0.521 0.477 0.669   | **0.994** **0.985** **0.994** |
> | GCFAgg (23 CVPR)    | 0.781 0.847 0.836   | 0.275 0.263 0.361   | 0.521 0.429 0.615   | 0.990 0.974 0.990   |
> | SEM (24 NeurIPS)    | 0.995 0.984 0.995   | 0.318 0.309 0.404   | 0.533 0.584 0.716   | **0.994** 0.983 **0.994** |
> | MVCAN (24 CVPR)     | 0.933 0.861 0.933   | 0.244 0.244 0.341   | 0.562 0.478 0.663   | 0.856 0.840 0.856   |
> | SCMVC (24 TMM)      | **0.996** **0.988** **0.996** | 0.291 0.277 0.376   | 0.562 0.420 0.627   | **0.994** **0.985** **0.994** |
> | DDMVC (25 PR)       | 0.957 0.893 0.957   | --  -- --               | 0.456 0.346 0.592   | 0.931 0.892 0.931   |
> | SSLNMVC (25 TMM)    | 0.995 0.985 0.995   | 0.235 0.244 0.430   | 0.521 0.510 0.686   | **0.994** 0.984 **0.994** |
> | AICN-MLM (25 AAAI)  | 0.990 0.971 0.990   | -- -- --               | 0.538 0.472 0.675   | **0.994** **0.985** **0.994** |
> | DFL-NET (25 TKDE)   | 0.995 0.983 0.995   | 0.227 0.246 0.600   | **0.586** 0.572 0.728 | 0.993 0.982 0.993   |
> | ours            | **0.996** *0.986* **0.996** | **0.356** **0.332** **0.445** | *0.574* **0.599** **0.775** | **0.994** *0.984* **0.994** |

---

> ### Author Response · Authors · 2025-11-27
>
> **Response to question 3:**
>
> We greatly appreciate your valuable suggestions and have further clarified the core distinctions between the RAV and SEM methods in the paper. The core of the SEM method lies in introducing adaptive weighted contrast learning at the feature layer. However, this approach still faces fundamental limitations. First, SEM primarily focuses on adjusting contrast targets at the feature layer, while the relational structure of samples is crucial for maintaining cross-view consistent neighborhood relationships. Neglecting consistency in the cross-view relational matrix may distort the local topological structure of samples, undermining semantic reliability and the accurate measurement of view similarity. Second, although SEM employs a similarity-based weighting strategy, it is confined to the feature layer without extending to the label layer. Consequently, this approach may still force low-similarity views toward consistency, potentially triggering semantic conflicts and disrupting inter-view relationships. In contrast, our framework emphasizes the tight integration and complementarity of relationship alignment and view-aware adaptive weighting. It not only ensures consistency of sample relationships at the structural level but also achieves adaptive alignment at the semantic level. It strengthens semantic consistency for highly similar views while reducing alignment strength for low-similarity views, thereby fundamentally avoiding semantic conflicts caused by forced alignment.
>
> **Response to question 4:**
> We appreciate the insightful questions raised by the reviewers. Our choice of Wasserstein distance (WD) as the view similarity metric stems primarily from its unique advantages in handling distributional differences. Specifically, the WD originates from optimal transport theory and quantifies the minimal cost required to transfer one distribution to another. This geometric property enables WD to provide meaningful distance metrics even when feature distributions between two views do not overlap, thereby delivering more stable and reliable weights for our adaptive weighting mechanism. In contrast, when distributions do not overlap, the MMD metric may be insufficiently sensitive to differences in distributions, while cosine similarity is sensitive to the direction of sample vectors and is better suited for measuring sample consistency but insensitive to differences in overall distributions. Within our RAV framework, the WD-based dynamic weights accurately reflect the intrinsic differences between views: strengthen label alignment for view pairs with smaller WD values, while weakening it for pairs with larger WD values. This approach effectively avoids semantic conflicts arising from forced alignment. Experimental results demonstrate that the selection of the WD metric enables our framework to achieve robust and efficient clustering across diverse data scenarios, validating the effectiveness and applicability of this approach.
>
> Different weighting strategies $W$ on the NGs, Digit-Product, ALOI, and NoisyMNIST datasets
>
> | Datasets         | NGs                | Digit-Product      | ALOI               | NoisyMNIST         |
> | :-------------- | :-----------------: | :-----------------: | :-----------------: | :-----------------: |
> | Evaluation metrics| ACC NMI PUR    | ACC NMI PUR        | ACC NMI PUR        | ACC NMI PUR        |
> | ours w/ $W_{Cosine}$ | 0.922 0.799 0.922 | 0.995 0.984 0.995 | 0.785 0.895 0.796 | 0.984 0.954 0.984 |
> | ours w/ $W_{MMD}$    | 0.968 0.904 0.968 | **0.998** 0.992 **0.998** | 0.796 0.905 0.799 | **0.996** **0.987** **0.996** |
> | ours w/ $W_{WD}$     | **0.980** **0.934** **0.980** | **0.998** **0.993** **0.998** | **0.837** **0.911** **0.846** | **0.996** 0.986 **0.996** |
>
> **Reference:**
>
> [1] Song Wu, Yan Zheng, Yazhou Ren, Jing He, Xiaorong Pu, Shudong Huang, Zhifeng Hao, and Lifang He. Self-weighted contrastive fusion for deep multi-view clustering.
>
> [2] Zhe Chen, Xiao-Jun Wu, Tianyang Xu, and Josef Kittler. Dfl-net: Disentangled feature learning network for multi-view clustering. IEEE Transactions on Knowledge and Data Engineering, 2025b.EE Transactions on Multimedia, 26:9150–9162, 2024.

---

### Official Review · Reviewer_jE16 · 2025-10-23

**Soundness:** 3
**Presentation:** 3
**Contribution:** 3
**Rating:** 8
**Confidence:** 5

**Summary:**

This paper proposes a multi-view clustering method that combines relational alignment and view-aware weighting, addressing two core challenges in the field: insufficient preservation of sample neighborhood structure and adaptive utilization of view similarity. The experimental design is rigorous and logically clear, providing empirical support for the effectiveness of the proposed method.

**Strengths:**

1. It clearly points out the shortcomings of existing methods in preserving sample neighborhood structure and adaptively utilizing view similarity.
2. The relational alignment module and the view-aware weighting mechanism are ingeniously designed, and their combination exhibits strong synergistic effects.
3. The experimental design of this paper is well-structured and thoroughly validated.
4. It breaks away from the traditional feature alignment approach, shifting towards deeper structural relationship alignment. By introducing distributional distance awareness in contrastive learning, it avoids the semantic disruption caused by forced alignment.

**Weaknesses:**

1. There is an issue with inconsistent formatting of mathematical symbols, which should be carefully reviewed to enhance the reader's understanding of the work.
2. The discussion of future prospects in the conclusion section is not sufficiently detailed.
3. The description of the innovations is not particularly prominent.

**Questions:**

1. The paper introduces view-aware adaptive weighting, particularly the Wasserstein distance-based view similarity measurement. Please provide a detailed explanation of the advantages of Wasserstein distance in handling distributional differences.
2. The paper lacks visualizations of the feature space or clustering results, such as t-SNE plots, making it difficult to intuitively appreciate the improvements achieved by the proposed method in representation learning.
3. The formatting of mathematical symbols in the paper requires careful attention, such as \mathbf{q}_{ij}^v.
4. The conclusion summarizes the key points of the paper and outlines future research directions. Please further elaborate on its research value for future work from both theoretical and practical perspectives.
5. The current abstract is clear in its writing but somewhat flat in expression. It does not fully highlight the innovation of the research. It is recommended to make the core contributions more prominent and to clearly explain the inherent connection between the two innovative points.

---

> ### Author Response · Authors · 2025-11-27
>
> **Response to weakness 1 and question 3:**
>
> We sincerely appreciate the reviewers' valuable comments. We fully agree that the standardization and rigor of mathematical symbols and formulas are crucial for readability. Therefore, we have conducted a comprehensive review and revision of all symbols and equations in the manuscript. Key revisions include standardizing the formatting of vectors, matrices, and scalars, ensuring consistent placement of all subscripts and superscripts, unifying operators, and correcting equations. These modifications aim to enhance the paper's rigor, professionalism and readability, thereby allowing greater focus on our work. We extend our gratitude once again for your valuable feedback.
>
> **Response to weakness 2 and question 4:**
>
> We appreciate your constructive feedback. Based on your suggestions, we have expanded and discussed future research directions in the conclusion section. Specifically, theoretically, we will explore more robust relational structures and universal similarity measurement methods to enhance the model's performance across diverse scenarios. In practice, we will advance the application of this framework in complex scenarios involving view missingness and noisy interference, striving to improve its practical value and universality in real-world environments.
>
> **Response to weakness 3:**
>
> We sincerely appreciate your valuable feedback. Following your suggestions, we have revised several key sections of the paper, including the abstract, introduction, related work, and conclusion to more clearly highlight the core innovations of our work. Through these revisions, we believe the paper now more accurately and powerfully conveys the uniqueness of our approach, its distinctions from existing work, and its contribution to advancing the field of multi-view clustering. We anticipate these enhancements will elevate the paper's overall expressiveness, making it more persuasive and impactful.
>
> **Response to question 1:**
>
> We appreciate your insightful comments. Our choice of Wasserstein distance as a metric for view similarity stems from its robust theoretical foundation and superior performance in handling complex distribution differences, inspired by the pioneering work of [1]. Derived from optimal transport theory, the Wasserstein distance quantifies the minimal cost required to transport one distribution to another. It provides a meaningful distance metric even when feature distributions across views overlap minimally or not at all. Within our RAV framework, weights calculated based on Wasserstein distance reliably reflect genuine differences between views, thereby robustly modulating the intensity of label contrastive learning. This directly mitigates representation degradation caused by excessively low view similarity. Ablation studies (as shown in Table 6) further validate its effectiveness. Through these designs, we better capture subtle differences between diverse views, enhancing model robustness and performance.
>
> Table 6: Ablation study on the view-aware adaptive weighting mechanism for NGs, Digit-Product, ALOI and Cora datasets.
>
> | Datasets      | NGs                | Digit-Product      | ALOI               | Cora               |
> | :------------ | :-----------------: | :-----------------: | :-----------------: | :-----------------: |
> | Evaluation Metrics  | ACC NMI PUR   | ACC NMI PUR   | ACC NMI PUR   | ACC NMI PUR   |
> | ours w/o W    | 0.966 0.895 0.966  | **0.998** **0.993** **0.998** | 0.801 0.903 0.807  | 0.585 0.393 0.585  |
> | ours          | **0.980** **0.934** **0.980** | **0.998** **0.993** **0.998** | **0.826** **0.912** **0.830** | **0.592** **0.404** **0.598** |
>
> **Response to question 2:**
>
> We sincerely appreciate your valuable suggestions. We have supplemented the visualization analysis as requested to more intuitively demonstrate the improvements of the proposed method in representation learning. Specifically, we performed t-SNE visualization on the global feature $Z$ learned by the model on the Digit-Product dataset. As shown in Figure 3 in the paper, the cluster structures across different categories gradually become distinct as training progresses: inter-cluster distances increase while intra-clusters become increasingly compact. This visualization intuitively demonstrates that our method effectively learns discriminative and clustering-friendly feature representations through relational alignment and view-aware adaptive weight contrastive learning, providing visual support for the enhanced clustering performance.

---

> ### Author Response · Authors · 2025-11-27
>
> **Response to question 5:**
>
> We appreciate your valuable feedback. We have revised the abstract to more clearly highlight the core innovations of our research and their intrinsic connections. The revised abstract explicitly identifies the globally guided sample relationship alignment module and the view-aware adaptive label contrast strategy, while explaining the close relationship between these two modules. The former ensures consistency in the relationship structure, providing a reliable foundation for the latter to achieve robust semantic consistency at the clustering level. We believe the revised abstract more effectively highlights the innovation of our work and the logical connections between its components, thereby providing readers with a clearer research framework and contributions.
>
> **References:**
>
> [1] Jian Shen, Yanru Qu, Weinan Zhang, and Yong Yu. Wasserstein distance guided representation learning for domain adaptation. In Proceedings of the AAAI conference on artificial intelligence, volume 32, 2018.

---

### Official Review · Reviewer_Jhbm · 2025-10-27

**Soundness:** 3
**Presentation:** 3
**Contribution:** 3
**Rating:** 6
**Confidence:** 4

**Summary:**

This paper proposes two key core modules: first, aligning global and local relation matrices to maintain neighborhood consistency; second, dynamically adjusting the weights of view pairs based on WD to alleviate representation conflicts. Meanwhile, the experimental section thoroughly validates the effectiveness of the proposed method.

**Strengths:**

1.This paper not only considers the structural relationships between samples but also employs a more universal metric to assess the similarity between views.

2.This paper uses three major clustering evaluation metrics: ACC, NMI, and PUR. The experimental results are presented clearly, and the performance is highly convincing.

3.The conclusion not only summarizes the core aspects of the method but also offers suggestions for future research directions.

**Weaknesses:**

1.This paper does not provide the rationale for selecting the contrastive learning temperature parameters $τ_F$ and $τ_L$.

2.Algorithm 1 provides an overview of the training process but lacks details of the key steps.

3.The descriptions in the introduction are somewhat repetitive and redundant.

**Questions:**

1.This paper concatenates the features of all views to obtain $Z∈R^{N×V_d}$, and further constructs the global relation matrix. Compared to other fusion strategies (such as weighted fusion of features to construct the relation matrix), what are the advantages of choosing this strategy?

2.Does constructing the relation matrix lead to an increase in computational cost, creating certain limitations?

3.The weighting mechanism only applies to the label contrastive loss, while in the relation alignment module, all views are equally aligned with the global matrix. Could this be considered an inconsistent treatment?

4.Some descriptions in the paper are somewhat colloquial. It is recommended to revise them into a more formal and academic expression.

5.The weight between views is calculated as $1/2\(w_{v,u}+w_{u,v}\)$ in equation (12). Please explain why this representation is used.

6.Algorithm 1 is described concisely but lacks key details, such as how the relation matrix is obtained.

---

> ### Author Response · Authors · 2025-11-27
>
> **Response to weakness 1:**
> We sincerely appreciate your valuable suggestions. $τ_F$  and $τ_L$  are temperature parameters that control the relationship alignment and the label contrastive learning, respectively. Temperature parameters primarily influence the similarity distribution and adjust the weights of positive and negative samples. Our primary focus is on learning the relationship alignment and adaptive weight contrastive learning strategy, and we have not optimized the temperature parameters. We set $τ_F$  and $τ_L$  to 0.5 based on the following considerations: setting the temperature value to 0.5 provides balanced similarity without introducing additional complexity, while also aligning with the default settings of most baseline methods.
>
> **Response to weakness 2 and question 6:**
>
> Thank you for your valuable suggestions. We acknowledge that there were clarity issues in the original description of Algorithm 1. In the revised version, we have made modifications to the algorithm, particularly by adding details about the construction process of the specific view relationship matrix $S^v$ and the global relationship matrix $S$, as well as clarifying the specific training process in batches. These improvements are intended to help readers better understand the overall training process of our method and to highlight how our approach addresses the issues of large-scale data learning.
>
> **Response to weakness 3 and question 4:**
>
> Thank you for your constructive feedback. Based on your valuable suggestions, we thoroughly revise the paper to ensure more concise and accurate expression that meets academic standards. The main revisions include adding the inherent connections between the two core modules in the abstract and providing a deeper summary and future outlook in the conclusion. In the introduction, we include recent related research methods and redefine the distinctions between recent works and our approach. Additionally, we optimize Figure 1 to enhance the clarity and logical flow between the modules.
>
> **Response to question 1:**
>
> Thank you for your valuable feedback. We chose to use feature concatenation rather than weighted fusion to construct the global feature $Z$ based on the overall objectives of our method, aiming to better achieve robust sample relationship alignment and view-aware adaptive contrastive learning. Specifically, by directly concatenating the features $Z^1$,$Z^2$,…,$Z^v$ from each view into $Z \in \mathbb{R}^{(N \times Vd)}$. We can fully preserve the feature information of each view, which allows us to capture the complex relationships between samples across different views more effectively when constructing the sample relationship matrix using Gaussian kernel. In contrast, the weighted fusion tends to lose some view information beneficial for relationship construction during the fusion process. Furthermore, weighted fusion requires determining the weight for each view, which is itself a challenging task. Introducing complex weight-learning module may also increase the model's complexity. Therefore, adopting the feature concatenation strategy as a direct and efficient feature integration method enables the construction of a more information-rich global relationship matrix $S$. It also provides a more reliable and representative alignment target for the local relationship matrix $S^v$ of each view.
>
> **Response to question 2:**
>
> Thank you for your valuable feedback. We acknowledge that constructing the sample relationship matrix does introduce additional computational overhead, primarily due to the calculation of similarities between large-scale sample pairs. However, we believe this issue can be effectively managed. Our method adopts a batch processing strategy. Instead of constructing the relationship matrix over the entire dataset with a time complexity of $\mathcal{O}(N^2)$, we compute the relationship matrix only within each batch during each iteration, with a complexity of $\mathcal{O}(n^2)$. During training across the entire dataset, the overall computational cost scales linearly with the dataset size $N$. This ensures that our model can efficiently handle large-scale datasets and can run stably even on the YoutubeVideo dataset, which contains over 100,000 samples. Therefore, while the relationship alignment module introduces some computational overhead, it does not fundamentally limit our method's applicability.

---

> ### Author Response · Authors · 2025-11-27
>
> Response to question 3:
> Thank you for your insightful feedback. We would like to provide the following explanation. Relationship alignment involves learning a consistent sample relationship structure across views. To achieve this, we choose to treat all views equally, aligning each view's relationship matrix with the global relationship matrix. This prevents the excessive suppression of views that have significant differences but contain unique and complementary information, ensuring the learning of a truly stable and representative relationship structure. By aligning the relationship matrices of all views with the global relationship matrix equally, we construct a stable and consistent structural relationship matrix, which provides a more reliable foundation for subsequent tasks. On the other hand, adaptive weighted label contrastive learning ensures consistency at the clustering semantic level. It strengthens label contrastive learning for view pairs with similar features to promote clustering consistency. For view pairs with significant differences, it reduces their weight to avoid potential semantic conflicts and representation degradation caused by forced alignment. The relationship alignment and adaptive weighted label contrastive learning correspond to two distinct objectives: structural relationships and clustering-level semantics. The two work synergistically to improve the model's clustering performance.
>
> **Response to question 5:**
>
> We appreciate the valuable feedback provided by the reviewers. Regarding the design of the symmetric average weight $\frac{1}{2}(w_{(u,v)} + w_{(v,u)})$ in equation (12), we provide the following explanation: As described in equation (11) of the paper, we compute the view similarity based on the Wasserstein distance between the view features $Z^v$ and $Z^u$, and obtain the weight $w_((u,v))$ by row-normalizing (softmax) the matrix $W$. Although the Wasserstein distance itself is symmetric $WD(Z^v,Z^u)=WD(Z^u,Z^v)$, the row normalization operation leads to $w_{(v,u)}≠w_{(u,v)}$. If we directly use $w_{(v,u)}$ as the contrastive learning weight for the view pair $(v,u)$ will result in inconsistent contrastive strength for the same view pair in both directions, thereby undermining symmetry and fairness in multi-view learning. Therefore, we adopt the symmetric average weight $\frac{1}{2}(w_{(u,v)} + w_{(v,u)})$. This method ensures equal weights for view pairs $(v,u)$ and $(u,v)$ in contrastive learning, aligning with the inherent bidirectional nature of the view relationships and enhancing the stability while rationality of the training process. This design is a key component of our view-aware contrastive mechanism, which helps maintain adaptive weighting across views without breaking the structural symmetry.

---

### Official Review · Reviewer_UBJs · 2025-11-06

**Soundness:** 4
**Presentation:** 4
**Contribution:** 3
**Rating:** 6
**Confidence:** 1

**Summary:**

This paper proposes two modules—relationship alignment and view-aware weighting—that preserve the neighborhood structure and alleviate the representation conflict caused by low view similarity, demonstrating a certain level of innovation.

**Strengths:**

1. The experiment compares multiple methods on different datasets, and the comparison methods are comprehensive.
2. The ablation study not only validates the effectiveness of each loss component but also further verifies the effectiveness of the view-aware adaptive weighting.
3. A view-aware adaptive weighting mechanism based on Wasserstein distance is introduced to dynamically adjust the weight of view pairs in label contrastive learning, avoiding the forced alignment of inconsistent views.
4. This paper focuses on sample neighborhood structure consistency rather than traditional sample-level consistency.

**Weaknesses:**

1. The description in the introduction of this paper has issues with insufficient coherence, and the logical flow needs improvement.
2. This paper does not clearly specify whether the contrastive method strictly follows the configuration in the original paper, lacking necessary details.
3. The conclusion section is highly repetitive of the abstract, failing to explore the core value of the innovations in depth, and lacks a thorough analysis and summary of the research findings.

**Questions:**

1. This paper introduces two innovative modules. Please explain and describe the inherent connection between these two modules.
2. Sections 2.1 and 2.2 still lack sufficient research and discussion on related work. Please further enrich this section.
3. This paper cites two studies related to view weighting in the related work section (Xu et al. (2023) and Wu et al. (2024)), yet the experimental comparison only includes a performance comparison with the SEM method. It is recommended to add a comparative experiment with the study by Wu et al. (2024).
4. This paper uses the Gaussian kernel function to compute the relationship matrix between samples. Please explain the reason for choosing the Gaussian kernel instead of using Euclidean distance-based methods to compute the relationship matrix.
5. When the similarity between two views is extremely low, the corresponding weight is significantly reduced, potentially leading to excessive suppression of the view pair's contribution and the loss of limited yet valuable discriminative information from that view. Does the proposed method suffer from this issue?
6. In the Implementation Details section, it is recommended to include the dimensionality settings of the encoder's hidden layers, and to specify whether a consistent network structure was used across all datasets.

---

> ### Author Response · Authors · 2025-11-27
>
> **Response to weakness 1:**
>
> Thank you for your valuable suggestions. Based on your feedback, we have thoroughly revised the paper. This revision primarily focused on enhancing the coherence and logical flow of the content. In the Introduction and Related Work sections, we have incorporated recent relevant literature to ensure broader coverage of the latest research developments in the field. Additionally, we have provided a clearer explanation of the relationships and distinctions between existing methods and our approach, thereby more accurately highlighting the contributions of our method.
>
> **Response to weakness 2 and question 6:**
>
> We appreciate the reviewer’s important questions regarding the experimental details and the fairness of baseline comparisons in our paper. We acknowledge that the original manuscript lacked sufficient description of the experimental settings for our method and the baseline methods. We have added the following explanation: We strictly adhered to the principle of fair comparison. All baseline methods are implemented using the parameters recommended in their original papers or optimized through parameter search to achieve optimal performance. For our proposed method, we employ a unified network architecture across all datasets. The encoder configuration is: Input-Fc500-Fc500-Fc2000-Fc512, and the decoder as a symmetric structure Fc512-Fc2000-Fc500-Fc500-Output. The hyperparameters $τ_F$ and $τ_L$ are set to 0.5. In terms of the experimental setup, we employ identical configurations for our method and all baseline methods: a learning rate of 0.0003, Adam optimizer, batch size of 256, with evaluations conducted after 200 training epochs.
>
> **Response to weakness 3:**
>
> Thank you for your valuable comment. We also recognize that the conclusion section should provide a more in-depth summary of the study's innovative contributions and core values. Based on your feedback, we have revised the conclusion. The revised conclusion aims to more clearly articulate the importance of relational alignment in preserving cross-view neighborhood structures, as well as the role of the view-aware adaptive weighting mechanism in mitigating representation degradation and semantic conflicts. Additionally, we have clarified the interrelationship between these two modules. Furthermore, we provide insights into how our approach may inspire future multi-view learning research, particularly in exploring more general similarity metric mechanisms and extending methods to handle incomplete or noisy views. Through these modifications, we strive to make the conclusion section more profound and insightful.
>
> **Response to question 1:**
>
> Thank you for your insightful comments. We acknowledge that the manuscript did not sufficiently clarify the intrinsic connection between the innovations of these two modules. We will add the following explanation: Although these two modules operate at different levels, they collectively form a complete process from structural consistency to clustering consistency. Specifically, the relationship alignment module preserves the neighborhood structures among samples across views, clearly revealing which samples should be close to each other and which should be distant. This stable neighborhood relationship provides a reliable foundation for enhancing the accuracy of inter-view similarity measurement and enabling semantic alignment at the clustering level. Consequently, the view-aware adaptive weighted label contrastive learning module achieves more robust clustering consistency while avoiding the disruption of existing good structural relationships due to over-forced alignment.
>
> **Response to question 2:**
>
> Thank you for your valuable suggestions. We acknowledge that the survey of related works in the manuscript was insufficient and have revised it according to your recommendations. In the revised version, we have added recent research findings to ensure the related work section more comprehensively reflects the latest advancements in the field. Additionally, we have further refined and clarified the key distinctions between our approach and existing related work, highlighting the contributions and significance of our methodology in this research.

---

> ### Author Response · Authors · 2025-11-27
>
> **Response to question 3:**
>
> Thank you for your professional advice. We have recognized the shortcomings in our experimental design and will incorporate improvements in the revised version. In the new experiments, we have included SCMVC [1], mentioned in the Related Work, alongside the latest method DFL-NET [2] for comparison and analysis. This will further enhance the comprehensiveness and persuasiveness of our experimental validation. The results are as follows:
>
> Table2: Clustering results of all methods on the NGs, Digit-Product, and ALOI datasets.
>
> | Datasets           |             NGs              |           Digit-Product          |              ALOI             |
> |--------------------|:----------------:|:----------------:|:----------------:|
> | Evaluation Metrics | ACC  NMI  PUR    | ACC  NMI  PUR    | ACC  NMI  PUR    |
> | MFLVC (22 CVPR)    | 0.932  0.825  0.932 | 0.991  0.976  0.991 | 0.435  0.786  0.435 |
> | GCFAgg (23 CVPR)   | 0.894  0.742  0.894 | 0.988  0.968  0.988 | 0.790  0.917  0.809 |
> | SEM (24 NeurIPS)   | 0.856  0.673  0.856 | 0.991  0.976  0.991 | 0.771  0.899  0.787 |
> | MVCAN (24 CVPR)    | 0.470  0.271  0.470 | 0.989  0.967  0.989 | **0.849**  **0.929**  **0.864** |
> | SCMVC (24 TMM)     | 0.826  0.638  0.826 | *0.995*  *0.986*  *0.995* | 0.818  *0.925*  *0.835* |
> | DDMVC (25 PR)      | --         --      --| 0.968  0.931  0.968 |0.796  0.907 0.813 |
> | SSLNMVC (25 TMM)   | *0.936*  *0.842*  *0.936* | 0.990  0.973  0.990 | 0.541  0.814  0.558 |
> | AICN-MLM (25 AAAI) | 0.912  0.774  0.912 | 0.991  0.976  0.991 | 0.788  0.906  0.800 |
> | DFL-NET (25 TKDE)  | 0.904  0.786  0.904 | **0.998**  **0.993**  **0.998** | 0.825  0.915  0.832 |
> | ours               | **0.980**  **0.934**  **0.980** | **0.998**  **0.993**  **0.998** | *0.826*  0.912  *0.830* |
>
> Table 3: Clustering results of all methods on the Cora, NUSWIDE, and Caltech-5V datasets.
>
> | Datasets      | Cora                 | NUSWIDE             | Caltech-5V          |
> | :------------ | :------------------- | :------------------ | :------------------ |
> | Evaluation Metrics  | ACC NMI PUR      | ACC NMI PUR    | ACC NMI PUR     |
> | MFLVC (22 CVPR)     | 0.268 0.111 0.377   | 0.624 0.338 0.624   | 0.867 0.781 0.867   |
> | GCFAgg (23 CVPR)    | 0.220 0.051 0.304   | 0.596 0.336 0.596   | 0.799 0.697 0.799   |
> | SEM (24 NeurIPS)    | 0.220 0.028 0.313   | 0.588 0.317 0.588   | *0.901* 0.834 *0.901*   |
> | MVCAN (24 CVPR)     | *0.567* *0.385* **0.640** | 0.572 0.290 0.572   | **0.919** **0.856** **0.919** |
> | SCMVC (24 TMM)      | 0.311 0.160 0.388   | 0.622 0.354 0.622   | 0.871 0.783 0.871   |
> | DDMVC (25 PR)       | 0.323 0.141 0.407   | 0.607 0.312 0.636   | 0.771 0.695 0.779   |
> | SSLNMVC (25 TMM)    | 0.277 0.102 0.364   | *0.637* *0.367* *0.637*   | 0.881 0.789 0.881   |
> | AICN-MLM (25 AAAI)  | 0.331 0.171 0.418   | 0.612 0.337 0.612   | 0.898 0.828 0.898   |
> | DFL-NET (25 TKDE)   | 0.359 0.185 0.435   | 0.614 0.325 0.614   | 0.833 0.767 0.833   |
> | ours            | **0.592** **0.404** *0.598* | **0.647** **0.371** **0.647** | *0.901* *0.839* *0.901*   |
>
> Table 4: Clustering results of all methods on the NoisyMNIST, YoutubeVideo, 3Sources, and Fashion datasets.
>
> | Datasets      | NoisyMNIST          | YoutubeVideo        | 3Sources            | Fashion             |
> | :------------ | :------------------ | :------------------: | :-----------------: | :------------------: |
> | Evaluation Metrics   | ACC NMI PUR    | ACC NMI PUR  | ACC NMI PUR   | ACC NMI PUR   |
> | MFLVC (22 CVPR)     | 0.988 0.965 0.988   | 0.238 0.224 0.324   | 0.521 0.477 0.669   | **0.994** **0.985** **0.994** |
> | GCFAgg (23 CVPR)    | 0.781 0.847 0.836   | 0.275 0.263 0.361   | 0.521 0.429 0.615   | 0.990 0.974 0.990   |
> | SEM (24 NeurIPS)    | 0.995 0.984 0.995   | 0.318 0.309 0.404   | 0.533 0.584 0.716   | **0.994** 0.983 **0.994** |
> | MVCAN (24 CVPR)     | 0.933 0.861 0.933   | 0.244 0.244 0.341   | 0.562 0.478 0.663   | 0.856 0.840 0.856   |
> | SCMVC (24 TMM)      | **0.996** **0.988** **0.996** | 0.291 0.277 0.376   | 0.562 0.420 0.627   | **0.994** **0.985** **0.994** |
> | DDMVC (25 PR)       | 0.957 0.893 0.957   | --  -- --               | 0.456 0.346 0.592   | 0.931 0.892 0.931   |
> | SSLNMVC (25 TMM)    | 0.995 0.985 0.995   | 0.235 0.244 0.430   | 0.521 0.510 0.686   | **0.994** 0.984 **0.994** |
> | AICN-MLM (25 AAAI)  | 0.990 0.971 0.990   | -- -- --               | 0.538 0.472 0.675   | **0.994** **0.985** **0.994** |
> | DFL-NET (25 TKDE)   | 0.995 0.983 0.995   | 0.227 0.246 0.600   | **0.586** 0.572 0.728 | 0.993 0.982 0.993   |
> | ours            | **0.996** *0.986* **0.996** | **0.356** **0.332** **0.445** | *0.574* **0.599** **0.775** | **0.994** *0.984* **0.994** |

---

> ### Author Response · Authors · 2025-11-27
>
> **Response to question 4:**
>
> Thank you for your professional comments. We indeed give thorough consideration to selecting an appropriate distance metric to capture structural relationships among samples. In this paper, we ultimately adopt the Gaussian kernel function $s_{ik}^{v} = \exp\left( -\frac{\||z_{i,:}^v - z_{k,:}^v \||^2}{\sigma} \right)$ to compute the relationship matrix between samples, rather than employing a Euclidean distance-based method commonly used to calculate sample relationships via $s_{ik}^{v} = \frac{1}{1 + \||z_{i,:}^v - z_{k,:}^v\||_2^2}$. This choice is primarily based on the following considerations: the Gaussian kernel function exhibits nonlinear exponential decay properties and effectively captures stable local neighborhood relationships crucial for clustering tasks while preserving local structure. Additionally, the adjustable parameter $\sigma$ of the Gaussian kernel enables adaptation to datasets with varying distributions and demonstrates strong robustness against noise and outliers. In our experiments, we set $\sigma$ to 1. In contrast, while methods based on the inverse of Euclidean distance are computationally simpler, it tends to suffer from poor numerical stability, disrupting the balance of the relationship matrix and failing to accurately reflect the true neighborhood relationships between samples. This can result in lower-quality relationship matrices. Therefore, we believe that employing Gaussian kernel functions enables the construction of higher-quality, more stable relationship matrices, laying a solid foundation for subsequent relationship alignment modules.
>
> **Response to question 5:**
>
> We appreciate the reviewers' detailed comments. Your concerns are well-founded, and we provide the following detailed explanation regarding the relevant issues: In our method, the adaptive weighting mechanism only reduces the weight of low-similarity view pairs during the label contrastive learning phase. While this somewhat weakens the intensity of contrastive learning, it does not diminish the informational contribution of the views themselves. All views remain fully engaged in reconstruction, relation alignment, and final label fusion, thus preserving their unique information. Conversely, forcibly aligning highly divergent views may induce semantic conflicts. The adaptive weighting mechanism is specifically designed to address such issues, avoiding forced alignment while preserving the unique information of each view to a certain extent. The results of the Ablation studies (Table 6) show that the performance improves after introducing adaptive weights, indicating that the effective information from the views is utilized more reasonably. In future research, we will explore more refined weighting strategies to balance consistency and complementary information better.
>
> Table 6: Ablation study on the view-aware adaptive weighting mechanism for NGs, Digit-Product, ALOI and Cora datasets.
>
> | Datasets      | NGs                | Digit-Product      | ALOI               | Cora               |
> | :------------ | :-----------------: | :-----------------: | :-----------------: | :-----------------: |
> | Evaluation Metrics  | ACC NMI PUR   | ACC NMI PUR   | ACC NMI PUR   | ACC NMI PUR   |
> | ours w/o W    | 0.966 0.895 0.966  | **0.998** **0.993** **0.998** | 0.801 0.903 0.807  | 0.585 0.393 0.585  |
> | ours          | **0.980** **0.934** **0.980** | **0.998** **0.993** **0.998** | **0.826** **0.912** **0.830** | **0.592** **0.404** **0.598** |
>
> **Reference:**
>
> [1] Song Wu, Yan Zheng, Yazhou Ren, Jing He, Xiaorong Pu, Shudong Huang, Zhifeng Hao, and Lifang He. Self-weighted contrastive fusion for deep multi-view clustering.
>
> [2] Zhe Chen, Xiao-Jun Wu, Tianyang Xu, and Josef Kittler. Dfl-net: Disentangled feature learning network for multi-view clustering. IEEE Transactions on Knowledge and Data Engineering, 2025b.EE Transactions on Multimedia, 26:9150–9162, 2024.

---

### Meta-Review · Area_Chair_jMNs · 2025-12-27

**Summary:**

RAV proposes two complementary components for deep multi-view clustering: relationship alignment, which aligns each view’ s sample-wise relation matrix to a global relation matrix to preserve cross-view neighborhood structure; and view-aware adaptive weighting. Given the positive scores from all the reviewers, I recommend accepting this paper.

**Reviewer Concerns:**

Most concerns focused on descriptive and formatted issues, which have been resolved well.

**Reviewer Scores:**

According to the author's response, the reviewers may maintain their positive scores.

---

### Decision · Program_Chairs · 2026-01-26

Accept (Poster)